# Development of a Monoclonal Antibody and a Serodiagnostic Lateral-Flow Device Specific to *Rhizopus arrhizus* (Syn. *R. oryzae*), the Principal Global Agent of Mucormycosis in Humans

**DOI:** 10.3390/jof8070756

**Published:** 2022-07-21

**Authors:** Genna E. Davies, Christopher R. Thornton

**Affiliations:** 1ISCA Diagnostics Ltd., B12A, Hatherly Laboratories, Prince of Wales Road, Exeter EX4 4PS, UK; g.davies@exeter.ac.uk; 2MRC Centre for Medical Mycology, Geoffrey Pope Building, University of Exeter, Stocker Road, Exeter EX4 4QD, UK

**Keywords:** mucormycosis, *Rhizopus*, monoclonal antibody, biomarker, lateral-flow device

## Abstract

Mucormycosis is a highly aggressive angio-invasive disease of humans caused by fungi in the zygomycete order, Mucorales. Though a number of different species can cause mucormycosis, the principal agent of the disease worldwide is *Rhizopus arrhizus*, which accounts for the majority of rhino-orbital-cerebral, pulmonary, and disseminated infections in immunocompromised individuals. It is also the main cause of life-threatening infections in patients with poorly controlled diabetes mellitus, and in corticosteroid-treated patients with SARS-CoV-2 infection, where it causes the newly described disease, COVID-19-associated mucormycosis (CAM). Diagnosis currently relies on non-specific CT, a lengthy and insensitive culture from invasive biopsy, and a time-consuming histopathology of tissue samples. At present, there are no rapid antigen tests for the disease that detect biomarkers of infection, and which allow point-of-care diagnosis. Here, we report the development of an IgG1 monoclonal antibody (mAb), KC9, which is specific to *Rhizopus arrhizus* var. *arrhizus* (syn. *Rhizopus oryzae*) and *Rhizopus arrhizus* var. *delemar* (*Rhizopus delemar*), and which binds to a 15 kDa extracellular polysaccharide (EPS) antigen secreted during hyphal growth of the pathogen. Using the mAb, we have developed a competitive lateral-flow device (LFD) that allows rapid (30 min) and sensitive (~50 ng/mL running buffer) detection of the EPS biomarker, and which is compatible with human serum (limit of detection of ~500 ng/mL) and bronchoalveolar lavage fluid (limit of detection of ~100 ng/mL). The LFD, therefore, provides a potential novel opportunity for the non-invasive detection of mucormycosis caused by *Rhizopus arrhizus*.

## 1. Introduction

Mucormycosis is a rare, but highly aggressive, angio-invasive disease of humans caused by fungi in the zygomycete order Mucorales, and is the second most important mould disease of humans after aspergillosis [1]. Of the more than 20 species of mucoralean fungi known to cause infections in humans [1], *Rhizopus arrhizus* (synonym, *Rhizopus oryzae*) is responsible for the majority of life-threatening infections worldwide in both paediatric and adult populations [2,3,4,5,6,7,8,9,10,11]. It accounts for ~90% of cases of rhino-orbital-cerebral mucormycosis (ROCM), especially in those with poorly controlled diabetes mellitus and ketoacidosis [5,9,10,12,13,14,15,16,17,18,19,20,21,22,23], but also in ostensibly immunocompetent individuals [24,25,26]. In addition, it is the leading cause (~70% of all cases) of pulmonary, gastrointestinal, cutaneous, and sub-cutaneous disseminated mucormycosis in immunocompromised individuals with haematological malignancies, solid organ and stem cell transplant recipients, and those receiving high-dose intravenous corticosteroids [7,9,26,27,28,29,30,31,32,33,34,35,36,37,38,39,40,41,42,43,44,45,46,47]. The fungus has emerged as the cause of necrotising super-infections in patients with severe influenza and with SARS-CoV-2 [46,48,49,50,51,52,53,54,55,56,57], and is a major contributor to the more than 50,000 cases and over 4000 deaths from COVID-19-associated mucormycosis (CAM) in India and elsewhere since the outbreak of the coronavirus pandemic in 2020 [57,58,59,60,61,62,63,64,65,66,67]. Many patients who have survived infections (known erroneously as black fungus disease due to the associated tissue necrosis) have been left with severe facial disfigurements or blindness due to soft tissue and bone damage following rhino-orbital infection, or as the result of the aggressive surgery needed to stem infections. 

Mucormycosis is associated with high rates of mortality, particularly in low- and middle-income countries [61,66,67,68], with an overall all-cause mortality rate of 54% [10], driven by slow diagnosis and delayed treatment with Mucorales-active antifungal drugs [65]. Given the current difficulties in the early detection of the disease [69], exacerbated by non-specific radiological indicators in computed tomography and magnetic resonance imaging, the insensitivity of culture from patient biopsy, the time-consuming and challenging nature of histopathology, and the lack of serological indicators of infection [66,70,71,72], a simple and rapid biomarker test for *R. arrhizus* infection is desirable. Lateral-flow technology is ideally suited to resource-limited settings [73], where the cost and complexity of more sophisticated diagnostic modalities for mucormycosis, such as MALDI-TOF [74] and PCR [reviewed in 71], hinder point-of-care detection of the disease. 

In this paper, we describe the development of a murine monoclonal antibody and a competitive lateral-flow device (LFD) specific to *Rhizopus arrhizus*, the principal global agent of mucormycosis in humans. We show that the test, when combined with a cube reader, has a limit of detection of ~50 ng *R. arrhizus* EPS/mL, and can be used to detect the biomarker in human serum and bronchoalveolar lavage fluid (BALf). This is the first time, to the best of our knowledge, that a mAb specific to *R. arrhizus* has been developed and used in a rapid point-of-care test (POCT) for the detection of this life-threatening pathogen. 

## 2. Materials and Methods

### 2.1. Ethics Statement

The hybridoma work described in this study was conducted under a UK Home Office Project Licence, and was reviewed by the institution’s Animal Welfare Ethical Review Board (AWERB) for approval. The work was carried out in accordance with The Animals (Scientific Procedures) Act 1986 Directive 2010/63/EU, and followed all the Codes of Practice which reinforce this law, including all elements of housing, care, and euthanasia of the animals. 

### 2.2. Fungal Culture

Fungi (Table 1) were routinely cultured on potato dextrose agar (PDA; P2182, Sigma, St. Louis, MO, USA). The medium was autoclaved 121 °C for 15 min prior to use, and fungi were grown at 30 °C or 37 °C under a 16 h fluorescent light regime to stimulate sporulation. To induce sporulation in *Apophysomyces* spp., the fungi were grown on autoclaved Czapek Dox agar (CDA; 70185, Sigma) at 37 °C. To induce the sporulation of *Saksenaea vasiformis*, the method of Padhye and Ajello [75] was used. 

### 2.3. Production of Hybridomas and Screening by ELISA 

Extracellular polysaccharides (EPS) were prepared using a proprietary purification method from culture filtrates of fungi grown for 6 d at 30 °C with shaking (100 rpm) in YNB + G medium (YNB; 51483, Sigma containing 3% (wt:vol) glucose) inoculated with 5 × 10^3^ spores/mL. For hybridoma production, the immunogen comprised a 1 mg/mL solution of EPS from *Rhizopus arrhizus* var. *arrhizus* (strain CBS112.07). Six-week-old BALB/c white mice were each given four intra-peritoneal injections (300 µL per injection) of immunogen at 2-wk intervals, and a single booster injection 5 d before fusion. Hybridoma cells were produced by the method described elsewhere [76], and monoclonal antibody (mAb)-producing clones were identified in indirect ELISA tests by using 20 μg EPS/mL phosphate-buffered saline (PBS; 137 mM NaCl, 2.7 mM KCl, 8 mM Na_2_HPO_4_, 1.5 mM KH_2_PO_4_ (pH 7.2)) immobilised to the wells of Maxisorp microtiter plates (Nunc) at 50 µL/well. The wells containing the immobilised antigen were incubated with 50 µL of mAb hybridoma tissue culture supernatant (TCS) for 1 h; after which, the wells were washed three times, for 5 min each, with PBST (PBS containing 0.05% (vol:vol) Tween-20). Goat anti-mouse polyvalent immunoglobulin (G, A, M) peroxidase conjugate (A0412, Sigma), diluted 1:1000 in PBST, was added to the wells and incubated for a further hour. The plates were washed with PBST as described, given a final 5 min wash with PBS, and bound antibody was visualised by incubating the wells with tetramethyl benzidine (TMB) substrate solution [76] for 30 min; after which, the reactions were stopped by the addition of 3 M H_2_SO_4_. Absorbance values were determined at 450 nm using a microplate reader (Tecan GENios, Tecan Austria GmbH, Grödig, Austria). Control wells were incubated with tissue culture medium (TCM) containing 10% (*v*/*v*) foetal bovine serum (FBS; FCS-SA, Biosera, Labtech International, Heathfield, UK) only. All incubation steps were performed at 23 °C in sealed plastic bags. The threshold for the detection of the antigen in ELISA was determined from control means (2 × TCM absorbance values). These values were consistently in the range of 0.050–0.100. Consequently, absorbance values ≥ 0.100 were considered as positive for the detection of the antigen.

### 2.4. Determination of Ig Class and Sub-Cloning Procedure 

The Ig class of mAbs was determined by using antigen-mediated ELISA [76]. The wells of microtiter plates coated with 20 μg EPS/mL PBS were incubated successively with hybridoma TCS for 1 h, followed by goat anti-mouse IgG1, IgG2a, IgG2b, IgG3, IgM, or IgA-specific antiserum (ISO-2, Merck Life Science UK Ltd., Gillingham, UK), diluted 1:3000 in PBST for 30 min; and rabbit anti-goat peroxidase conjugate (A5420, Sigma), diluted 1:1000 for a further 30 min. The bound antibody was visualised with TMB substrate as described. Hybridoma cell lines were sub-cloned three times by limiting dilution, and cell lines were grown in bulk in a non-selective medium, preserved by slowly freezing in FBS/dimethyl sulfoxide (92:8 vol:vol), and stored in liquid N_2_.

### 2.5. Production of Rabbit Antiserum 

Antiserum was generated in rabbits immunised with purified EPS from *R. arrhizus* var. *arrhizus* CBS112.07. The immunisations were carried out by Eurogentec (Seraing, Belgium) following an 87-d immunisation regimen, with animals immunised on days 0, 14, 28, and 56. Final bleeds were taken on day 87, and the serum was harvested for purification. 

### 2.6. Antibody Purification and Enzyme Conjugation 

The hybridoma TCS of mAb KC9 was harvested by centrifugation at 2147× *g* for 40 min at 4 °C, followed by filtration through a 0.8 μM cellulose acetate filter (10462240, GE Healthcare Life Sciences, Amersham, UK). The culture supernatant was loaded onto a HiTrap Protein A column (17-0402-01, GE Healthcare Life Sciences) using a peristaltic pump P-1 (18-1110-91, GE Healthcare Life Sciences) with a low pulsation flow of 1 mL/min. The columns were equilibrated with 10 mL of PBS, and the column-bound antibody was eluted with 5 mL of 0.1 M glycine-HCl buffer (pH 2.5) with a flow rate of 0.5 mL/min. The buffer of the purified antibody was exchanged to PBS using a disposable PD-10 desalting column (17-0851-01, GE Healthcare Life Sciences). Following purification, the antibody was sterile-filtered with a 0.24 µm syringe filter (85037-574-44, Sartorius UK Ltd., Epsom, UK), and stored at 4 °C. The rabbit antiserum, SK0078, was similarly purified using Protein G. Protein concentrations were determined using a NanoDrop spectrophotometer with the protein concentrations calculated using the mass extinction coefficient of 13.7 at 280 nm for a 1% (10 mg/mL) IgG solution. Antibody purity was confirmed by SDS-PAGE and gel staining using Coomassie Brilliant Blue R-250 dye (Thermo Fisher Scientific UK Ltd., Loughborough, UK). Protein-A-purified mAb KC9 or pAb SK0078 were conjugated to horseradish peroxidase (HRP) for ELISA studies using a Lightning-Link horseradish peroxidase conjugation kit (701-0000; Bio-Techne Ltd., Abingdon, UK), or to alkaline phosphatase (AKP) for western blotting studies using a Lightning-Link alkaline phosphatase conjugation kit (702-0010; Bio-Techne Ltd.). 

### 2.7. Antibody Specificity Tests

For antibody specificity tests, fungi were grown for 48 h at 30 °C in YNB + G liquid medium with shaking (100 rpm). The culture fluids were filtered through a Miracloth, and filtrates were double diluted in PBS in the wells of microtiter plates. The wells containing immobilised antigens were washed, dried, and assayed by direct ELISA using KC9-HRP and SK0078-HRP conjugates at 1:5000 and 1:1000, respectively. 

### 2.8. Epitope Characterisation by Heat and Periodate Oxidation 

The heat stability of the KC9 epitope was determined by heating EPS from the *R. arrhizus* var. *arrhizus* strain, CBS112.07, at a concentration of 20 μg/mL PBS in a boiling water bath. At 10 min intervals, 50 μL volumes were removed, and, after cooling, were transferred to the wells of microtiter plates for assay by direct ELISA using mAb KC9 conjugated to HRP (KC9-HRP) at a concentration of 1:5000 in PBST. For periodate oxidation, microtitre wells containing immobilised EPS at 20 μg/mL PBS were incubated with 50 μL of sodium *meta*-periodate solution (20 mM NaIO_4_ in 50 mM sodium acetate buffer (pH4.5)) or acetate buffer only (control) for 24, 4, 3, 2, 1, or 0 h at 4 °C in sealed plastic bags. The plates were given four 3-min PBS washes before processing by direct ELISA. 

### 2.9. Polyacrylamide Gel Electrophoresis and Western Blotting

Sodium-dodecyl-sulphate-polyacrylamide gel electrophoresis (SDS-PAGE) was carried out using 4–20% gradient polyacrylamide gels (161-1159, Bio-Rad, Hercules, CA, USA) under denaturing conditions. The antigens were separated electrophoretically at 165 V, and pre-stained markers (161-0318, Bio-Rad) were used for molecular weight determinations. For western blotting, the separated antigens were transferred electrophoretically onto a PVDF membrane (162-0175, Bio-Rad) for 2 h at 75 V, and the membrane was blocked for 16 h at 4 °C in PBS containing 1% (wt:vol) BSA. The blocked membranes were incubated with KC9-AKP or SK0078-AKP conjugates, diluted 1:15,000 or 1:5000, respectively, in PBS containing 0.5% (wt:vol) BSA (PBSA) for 2 h at 23 °C. The membranes were washed three times with PBS and once with PBST, and the bound antibody was visualised by incubation in the substrate solution [76]. The reactions were stopped by immersing membranes in dH_2_O, and the membranes were then air dried between sheets of Whatman filter paper.

### 2.10. Competitive Lateral-Flow Device

The competitive lateral-flow device (LFD) was manufactured by Lateral Dx (Alloa, Scotland, UK). The test consisted of a Kenosha 75 mm backing card; Ahlstrom 222 and 1281 top and sample pads, respectively; and a CN95 (12 μm) nitrocellulose membrane. The test (T) line consisted of EPS from the *R. arrhizus* var. *arrhizus* strain, CBS112.07, at a concentration of 0.2 mg/mL, whereas the internal test control (C) line consisted of goat anti-mouse IgG (Arista Biologicals) at a concentration of 0.25 mg/mL. 

### 2.11. LFD Specificity and Sensitivity

The specificity of the LFD was determined using running buffer (PBS containing 0.1% (vol:vol) Tween-20) containing 100 μg/mL of purified EPS prepared from human-pathogenic mucoralean fungi (*Apophysomyces variabilis* (strain CBS658.93), *Rhizopus arrhizus* var. *arrhizus* (strain CBS112.07), *Mucor circinelloides* (strain B5-2), *Cunninghamella bertholletiae* (strain CBS115.80), *Lichtheimia corymbifera* (strain CBS109940), *R*. *microsporus* var. *rhizopodiformis* (strain CBS102277), *Rhizopus oryzae* (strain CBS 111233), and *Rhizomucor pusillus* (strain CBS120587)). The experimental control consisted of running the buffer only. A volume of 100 μL of the sample was mixed with 4 μL (equivalent to 7.5 GU) of a 1.5 μg/mL solution of KC9 antibody conjugated to colloidal gold, and was incubated at 23 °C for 10 min. The solution was then added to the LFD, and the results recorded as negative (both C and T lines visible) or positive (C line only) after 30 min.

The analytical limit of detection (LOD) of the LFD was determined using purified EPS from the *R. arrhizus* var. *arrhizus* strain, CBS112.07, diluted in running buffer, with the running buffer only acting as the experimental control. A volume of 100 μL of the sample was incubated with KC9-gold conjugate, and, as described, was added to the LFD, and the T and C line intensities were recorded after 30 min on a scale of 0–10 using a score card or as artificial units (a.u.) using a cube reader.

### 2.12. LFD Serum and Bronchoalveolar Lavage Fluid Tests

#### 2.12.1. Spiked Serum

Normal serum from a healthy AB blood group male (Biosera) was spiked with purified EPS from the *R. arrhizus* var. *arrhizus* strain, CBS112.07, and was stored as aliquots at −20 °C prior to use. Upon thawing, 50 μL of spiked or control (unspiked) serum was mixed 1:2 (vol:vol) with PBS containing 0.5% Na_2_-EDTA, and was heated in a boiling water bath for 3 min. The heated mixture was centrifuged at 16,000× *g* for 5 min, the clear supernatant was mixed 1:1 (vol:vol) with PBS containing 0.2% (vol:vol) Tween-20, and the resultant 100 μL containing 80 μg/mL of EPS was incubated with KC9-gold conjugate as described. After 10 min, the solution was added to the LFD, and the test results were recorded as negative (both C and T lines visible) or positive (C line only) after 30 min. Separately, the LOD with spiked serum was determined using the cube reader, with normal (unspiked) serum acting as the control. 

#### 2.12.2. Spiked BALf

Normal BALf from a healthy 59-year-old male (BioIVT; HUMANBAL-0101312) was spiked with purified EPS from the *R. arrhizus* var. *arrhizus* strain, CBS112.07, and was stored as aliquots at −20 °C prior to use. Upon thawing, 50 μL spiked or control (unspiked) BALf was mixed 1:1 (vol:vol) with PBS containing 0.2% (vol:vol) Tween-20, and the resultant 100 μL containing 80 μg/mL of EPS was incubated with KC9-gold conjugate as described. After 10 min, the solution was added to the LFD, and the test results were recorded as negative (both C and T lines visible) or positive (C line only) after 30 min. Separately, the LOD with spiked BALf was determined using the cube reader, with normal (unspiked) BALf acting as the control.

### 2.13. Statistical Analysis

Numerical data were analysed using the statistical programme, Minitab (Minitab 16; Minitab, Coventry, UK). An analysis of variance (ANOVA) was used to compare means, and post hoc Tukey–Kramer analysis was then performed to determine the statistical significance.

## 3. Results

### 3.1. Production of Hybridomas and mAb Isotyping

Two hybridoma fusions were performed, and 686 hybridoma cell lines were tested in indirect ELISA for recognition of the immunogen. Forty cell-lines produced EPS-reactive antibodies, with all 40 producing mAbs of the immunoglobulin class, G1 (IgG1). 

### 3.2. Antibody Specificities 

A preliminary study of antigen production by the *R. arrhizus* var. *arrhizus* strain, CBS112.07, in YNB + G shake culture showed that the KC9 antigen was secreted into the culture medium, and that its production plateaued after 48 h, coincident with a cessation in hyphal growth of the pathogen (Appendix A). For this reason, the specificity of mAb KC9 was determined in western blotting (Figure 1) and direct ELISA (Table 1) studies using 48-h-old culture filtrates of fungi grown in a YNB + G shake culture. Unlike pAb SK0078, which reacts in western blots with antigens (molecular weights of between ~18 kDa to ~250 kDa) from all of the *Rhizopus* species tested (Figure 1A,B), mAb KC9 is specific to *Rhizopus arrhizus* var. *arrhizus*, *Rhizopus arrhizus* var. *delemar*, and *Rhizopus oryzae* (Figure 1C,D), binding to a single immuno-reactive band of ~15 kDa. Testing with mAb KC9 in direct ELISA against culture filtrates from other yeast and mould pathogens (Table 1) further demonstrated its species-specificity, with no cross-reaction of the mAb with related and unrelated human pathogens, including *Aspergillus* spp., *Candida albicans*, *Cryptococcus neoformans*, *Fusarium* spp., *Scedosporium* spp., and *Lomentospora prolificans*.

### 3.3. Epitope Characterisation

The epitope bound by mAb KC9 is heat-stable, with no significant effect on mAb binding when heating the EPS antigen at 100 °C for 60 min (Appendix A). The binding of mAb KC9 to its epitope is similarly insensitive to periodate oxidation (Appendix A). Taken together, this shows that the KC9 epitope is a heat-stable, periodate-insensitive carbohydrate moiety. 

### 3.4. Lateral-Flow Device 

#### 3.4.1. Specificity and Sensitivity

Using purified EPS from human-pathogenic mucoralean fungi, the LFD was shown to be species-specific, detecting *Rhizopus arrhizus* (syn. *R. oryzae*) only (Figure 2A). The species-specificity of mAb KC9 was further demonstrated by direct ELISA (Figure 2B) and western blot (Figure 2D) of the purified EPS preparations, with binding to the ~15 kDa antigen of *Rhizopus arrhizus* (*R. oryzae*) only. Unlike mAb KC9, pAb SK0078 reacted with all purified EPS preparations in both direct ELISA (Figure 2C) and western blot (Figure 2E), demonstrating the presence of immuno-reactive antigens of between ~18 kDa to ~250 kDa in all eight EPS preparations. 

The sensitivity of the LFD was determined using EPS from *R. arrhizus* var. *arrhizus* (CBS112.07) diluted into the running buffer. Using both a score card (Figure 3A) and a cube reader (Figure 3B), there were sequential and significant decreases in test (T) line intensities with increases in EPS concentrations between 0 μg EPS/mL (running buffer only) and 10 μg EPS/mL. Based on these results, the analytical limit of detection (LOD) was shown to be ~50 ng EPS/mL for the running buffer, using both scoring systems (Figure 3C,D).

#### 3.4.2. LFD Serum and Bronchoalveolar Lavage Tests

The LFD is compatible with human serum and BALf (Figure 3E). Though serum required a quick and simple sample pre-treatment step with heat/EDTA prior to incubation with the running buffer, BALf could be mixed directly with the running buffer for incubation and addition to the test. Using the cube reader, the LOD with serum was determined to be ~500 ng/mL, whereas the LOD with BALf was ~100 ng/mL. 

## 4. Discussion

In this paper, we describe the development and characterisation of a murine IgG1 monoclonal antibody (mAb), KC9, raised against an extracellular polysaccharide (EPS) antigen from *Rhizopus arrhizus* var. *arrhizus* (formerly *Rhizopus oryzae*), and the detection of the EPS biomarker using lateral-flow technology. 

Though mAbs and rabbit antiserum have previously been developed against immunodominant antigens of Mucorales [77,78,79], the intracellular nature of the antigens limits their use to the immunohistochemistry of infected tissues [79]. For point-of-care diagnostics employing lateral-flow technology, extracellular antigens are needed that act as circulating biomarkers of infection [1]. Ideally, these should be produced during the active growth of a pathogen, and the target epitope should be sufficiently robust to allow the pre-treatment of bodily fluids, such as serum or BALf. Heat-stable carbohydrate (polysaccharide) antigens are ideal for this purpose, and form the basis of lateral-flow assays and enzyme-linked immunosorbent assays (ELISA) for the detection of invasive pulmonary aspergillosis [1]. The species-specific mAb KC9 described here binds to a heat-stable EPS antigen produced during the active growth of the pathogen, and, therefore, potentially during angio-invasive growth in humans. The ability of the target antigen to withstand treatment with heat and EDTA treatment makes it well-suited to serum- or BALf-based diagnosis of *R. arrhizus*. To this end, we have incorporated the mAb into a lateral-flow device (LFD), which, when combined with a simple and well-established sample pre-treatment step, can be used to detect the diagnostic signature molecule in human serum and BALf. 

The current detection of infectious mucoralean fungi relies on sophisticated laboratory tests, including MALDI-TOF [74], PCR [reviewed in 71], or enzyme-linked immunospot (ELISpot) tests that detect Mucorales-specific T cells [80]. Though a 23 kDa *R. arrhizus*-specific protein has been detected in the serum of *R. arrhizus*-infected mice using polyclonal antibody-based ELISA [81,82], no mAb-based serodiagnostic lateral-flow tests currently exist for the specific detection of *R. arrhizus*. A mAb (2DA6) and a lateral-flow immunoassay (LFIA) have been developed that recognise *Rhizopus oryzae*, but the mAb lacks specificity, cross-reacting with an epitope on α-1,6 mannans conserved among human pathogenic yeasts and filamentous fungi, including *Candida albicans* and the angio-invasive moulds, *Aspergillus*, *Fusarium*, and *Scedosporium* [83]. Despite this, the LFIA was able to detect cell wall fucomannan in BALf, serum, and urine samples from diabetic ketoacidotic and neutropenic mice following intratracheal challenge with *Rhizopus delemar*, *Lichtheimia corymbifera*, *Mucor circinelloides*, and *Cunninghamella bertholletiae*, demonstrating the utility of carbohydrate biomarkers in the diagnosis of mucormycosis [84]. 

Cross-reactivity with other pathogenic moulds and yeasts is undesirable, especially in the setting of co-infections comprising *R. arrhizus* and *Aspergillus*, *Exophiala*, and *Fusarium* species [85,86,87,88,89], where discrimination of the infecting species is needed to optimise treatment with antifungal drugs, and to prevent breakthrough *R. arrhizus* infections [70,90]. The detection of mucormycosis is not possible using the pan-fungal (1→3)-β-d-glucan (BDG) test, since the Mucorales lack this carbohydrate in their cell walls [91]. However, it can be used to rule out invasive pulmonary aspergillosis [66], the most frequent differential diagnosis associated with mucormycosis [92]. When combined with the BDG test and more-specific immunoassays, such as the *Aspergillus* LFD and ELISA tests [1,91,93], the *R. arrhizus*-specific LFD described here might provide a useful and novel addition to the armamentaria needed for differential diagnosis of the first (aspergillosis) and second (mucormycosis) most common mould diseases of humans [92]. 

The *R. arrhizus* LFD is a competitive immunoassay which relies on a soluble antigen present in the patient sample (for example, serum and BALf), displacing binding of the gold-conjugated KC9 mAb to purified EPS present in the test line. The response is, therefore, negatively correlated to the analyte concentration (i.e., more analyte present, less signal; no analyte gives the highest signal). Competitive lateral-flow tests have found widespread applicability in medicine for the detection of cancer biomarkers and therapeutic drugs [94,95], in the detection of food- and water-borne pesticides and toxins [96,97], and in agriculture for the detection of plant pathogenic fungi [98]. The competitive format is ideally suited to low molecular weight antigens that possess a single antigenic determinant (epitope) for antibody binding. We chose the competitive LFD format, since we were unable to develop a sandwich LFD format using KC9 as both capture and detector species, or when used in combination with the rabbit antiserum, SK0078 (results not shown), indicating single epitope binding on the EPS antigen by mAb KC9. 

In the competitive LFD format, mAb KC9 retained the species-specificity displayed in ELISA and western blotting studies, binding to EPS from *R. arrhizus*, but not to EPS from other related and unrelated Mucorales of clinical relevance [99]. Though sandwich LFD formats usually show a higher analytical sensitivity (picograms of analyte per mL) compared to the competitive format (nanograms per mL), an advantage of the competitive format is absence of false negative results associated with the ‘high-dose hook effect’ seen in sandwich tests [100]. The competitive LFD reported here has an analytical limit of detection (LOD) of ~50 ng EPS/mL of the running buffer, determined both by visual assessment using a score card and also using a cube reader. The use of the cube reader removes the subjective visual appraisal of test positivity by the operator, providing a simple digital readout. The importance of a digital readout has recently been demonstrated with the IMMY *Aspergillus* GM LFA, where visual appraisal of the GM LFA test result can lead to significant numbers of false-positive results, impacting the test specificity [101,102]. In the absence of widespread testing of the LFD, we are not able, at this stage, to determine the clinical relevance of the LOD of 50 ng/mL with the running buffer, ~500 ng/mL with serum, and ~100 ng/mL with BALf, even though these concentrations of antigens are similar to those reported in cattle with experimental systemic bovine zygomycosis [103], and are comparable to the sensitivities of sandwich LFDs for the detection of *Aspergillus* and *Scedosporium* carbohydrate antigens [1,104]. The test, therefore, requires validation in the clinic to determine its diagnostic utility in human disease detection. However, we have shown that the test is capable of detecting the diagnostic EPS biomarker in both human serum and human BALf. Furthermore, due to the heat stability of the KC9 antigen and epitope, we were able to employ a standardised serum pre-treatment step (EDTA and boiling) also used in the *Aspergillus* LFD test for serum and BALf testing [105], providing an opportunity to use the same treated sample on two different LFD platforms. 

A disadvantage of the LFD is its inability to detect Mucorales other than *R. arrhizus*, such as *Lichtheimia* species, which are the second most important cause of mucormycosis in Europe after *R. arrhizus* [11,106,107] or *Apophysomyces* species and *Rhizopus microsporus*, which, alongside *R. arrhizus*, are important causes of COVID-19-associated mucormycosis [8,9,21,65,108,109]. To negate this, we have developed an *Apophysomyces*-specific mAb (JD4) and a pan-Mucorales-specific mAb (TG11) for incorporation into a multiplex LFD alongside KC9.

## 5. Trademark

The word, ZygoDx^®^ (EU018696066 (pending)), is protected by ISCA Diagnostics Ltd. through the European Union Intellectual Property Office (EUIPO).

## Figures and Tables

**Figure 1 jof-08-00756-f001:**
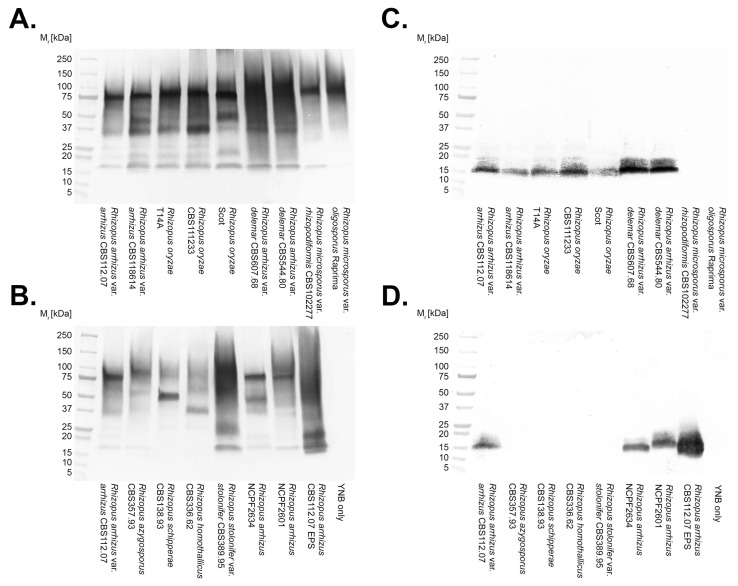
Western blots of culture filtrates from *Rhizopus* species using pAb SK0078 (**A**,**B**) and mAb KC9 (**C**,**D**). Though pAb SK0078 binds to antigens with molecular weights of between ~18 kDa to 250 kDa from all *Rhizopus* spp., mAb KC9 reacts with a single antigen of ~15 kDa, and is species-specific, reacting with different strains of *Rhizopus arrhizus* var. *arrhizus*, *Rhizopus arrhizus* var. *delemar*, and *Rhizopus oryzae* only. The positive control, comprising 20 μg of purified EPS from the *R*. *arrhizus* var. *arrhizus* strain, CBS112.07 (**B**,**D**), also yields a single KC9-reactive band of ~15 kDa, whereas the negative control, comprising YNB culture medium only, is negative both for pAbSK0078 and for mAb KC9.

**Figure 2 jof-08-00756-f002:**
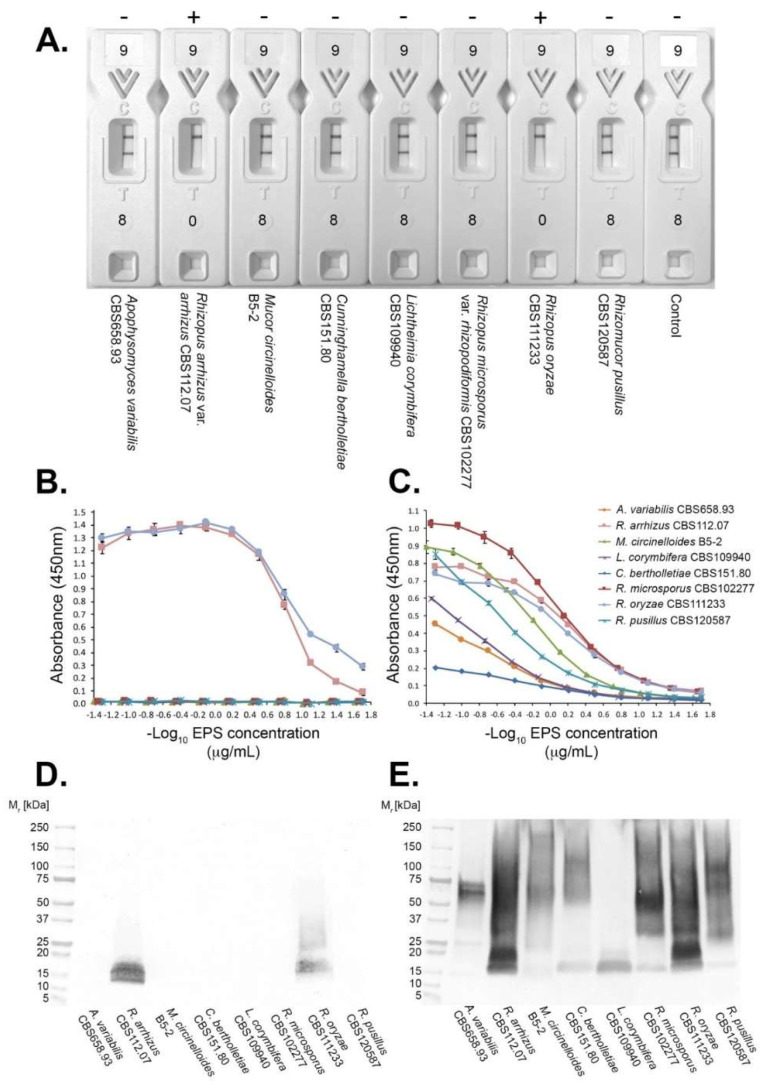
Specificity of the LFD. (**A**) Specificity of the LFD using 100 μg purified EPS/mL running buffer of the human-pathogenic mucoralean fungi, *Apophysomyces variabilis* (strain CBS658.93), *Rhizopus arrhizus* var. *arrhizus* (strain CBS112.07), *Mucor circinelloides* (strain B5-2), *Cunninghamella bertholletiae* (strain CBS115.80), *Lichtheimia corymbifera* (strain CBS109940), *R. microsporus* var. *rhizopodiformis* (strain CBS102277), *Rhizopus oryzae* (strain CBS 111233), and *Rhizomucor pusillus* (strain CBS120587). Species other than *R. arrhizus* var. *arrhizus* and *R. oryzae* had T lines similar to the control (running buffer only). EPS from *R. arrhizus* var. *arrhizus* and *R. oryzae* resulted in complete displacement of KC9-gold conjugate binding to the T line, demonstrating the species-specificity of the LFD. + indicates a positive test result, − indicates a negative test result. (**B**,**C**) ELISA of the purified EPS samples, showing specific binding of mAb KC9 to *R. arrhizus* var. *arrhizus* and *R. oryzae* (**B**), and broad reactivity of pAb SK0078 with all species (**C**). Each point is the mean of three replicates ± SE, and the threshold absorbance value for detection of antigen in ELISA is ≥0.100. (**D**,**E**) Western blots of the purified EPS samples, showing species-specific binding of mAb KC9 to an ~15 kDa antigen of *R. arrhizus* and *R. oryzae* (**D**), and the presence of pAb SK0078-reactive antigens (~15 kDa to ~250 kDa) in all samples (**E**). Each well contains 20 μg EPS.

**Figure 3 jof-08-00756-f003:**
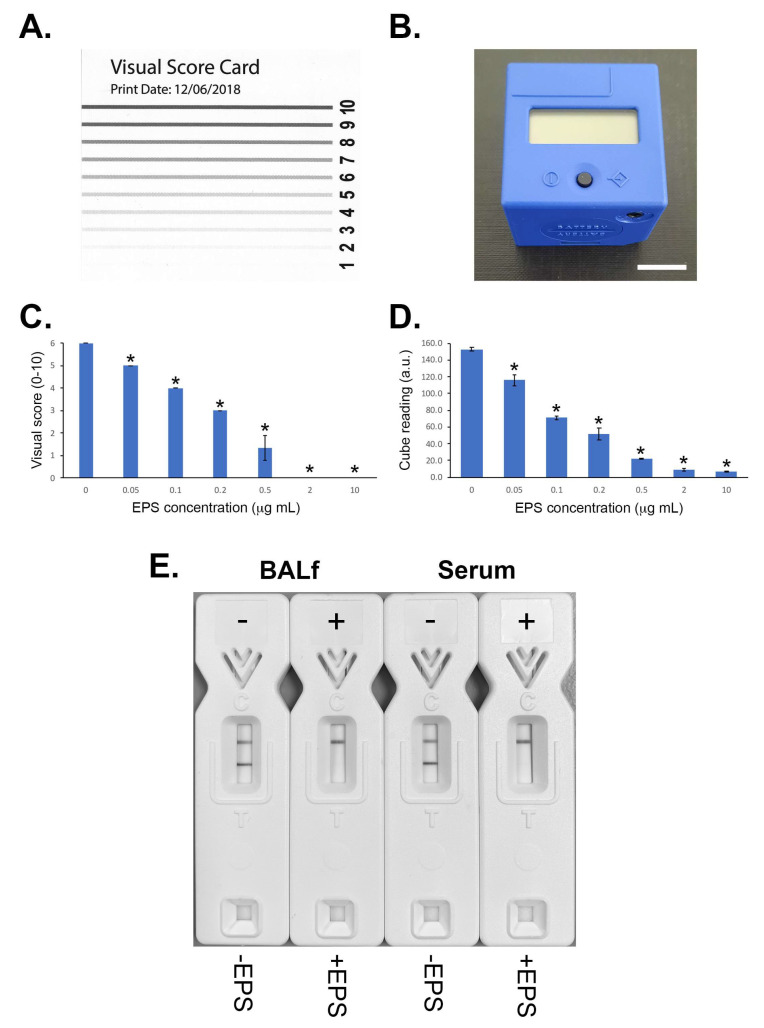
(**A**) Visual score card used for determinations of LFD test (T) and control (C) line intensities, recorded on a scale of 0–10. (**B**) Cube reader used for determination of T and C line intensities, recorded as artificial units (a.u.); scale bar = 1.5 cm. (**C**,**D**) Sensitivities of the LFD using the visual score card and cube reader systems, respectively. Bars are the means of three replicates ± 2 × SE, and * indicates a significant difference (Student’s *t*-test (*p* < 0.05) of mean values compared to the control (running buffer only)). All samples had control (**C**) line scores of 8 using the score card, and >300 a.u. using the cube reader. (**E**) Detection of the EPS biomarker in human BALf and serum. Samples were spiked with purified EPS from *R. arrhizus* var. *arrhizus* (CBS112.07) to give a final concentration of 80 μg/mL. Note the displacement of the T line with spiked BALf and serum samples, indicating a positive (+) test result. Normal (unspiked) BALf and serum samples gave a negative (−) test result (T lines present).

**Table 1 jof-08-00756-t001:** Details of fungi used in this study, and specificity of mAb KC9 in direct ELISA tests of 48-h-old culture filtrates of related and unrelated fungi.

Species	Isolate Number	Source ^1^	ELISA ^2^
*Absidia glauca*	2	CRT	0.060
*Absidia spinosa*	3	CRT	0.044
*Actinomucor elegans* var. *kuwaitensis*	117697	CBS	0.062
*Apophysomyces elegans*	477.78	CBS	0.058
*Apophysomyces mexicanus*	136361	CBS	0.039
*Apophysomyces ossiformis*	125533	CBS	0.037
*Apophysomyces variabilis*	658.93	CBS	0.067
*Aspergillus fumigatus*	Af293	FGSC	0.077
*Aspergillus flavus*	91856iii	IMI	0.055
*Aspergillus nidulans*	A4	FGSC	0.034
*Aspergillus niger*	102.4	CBS	0.033
*Aspergillus terreus* var. *terreus*	601.65	CBS	0.069
*Basidiobolus ranarum*	117.29	CBS	0.051
*Candida albicans*	SC5314	SB	0.058
*Cokeromyces recurvatus*	168.59	CBS	0.061
*Conidiobolus coronatus*	110.76	CBS	0.071
*Cryptococcus neoformans*	8710	CBS	0.070
*Cunninghamella bertholletiae*	151.8	CBS	0.030
*Fusarium oxysporum*	167.3	CBS	0.080
*Fusarium solani*	224.34	CBS	0.055
*Lichtheimia corymbifera*	109940	CBS	0.066
*Lichtheimia corymbifera*	120580	CBS	0.047
*Lichtheimia hyalospora*	146576	CBS	0.056
*Lichtheimia ornata*	142195	CBS	0.029
*Lichtheimia ramosa*	112528	CBS	0.088
*Lichtheimia ramosa*	124197	CBS	0.049
*Lichtheimia ramosa*	2845	NCPF	0.039
*Lomentospora prolificans*	3.1	CRT	0.062
*Mucor circinelloides*	E2A (FJ713065)	CRT	0.081
*Mucor circinelloides*	B5-2 (KT876701)	CRT	0.045
*Mucor indicus*	120.08	CBS	0.071
*Mucor mucedo*	95	CRT	0.056
*Mucor piriformis*	169.25	CBS	0.070
*Mucor plumbeus*	96	CRT	0.042
*Mucor racemosus* f. *racemosus*	111557	CBS	0.033
*Mucor racemosus* f. *racemosus*	112382	CBS	0.067
*Mucor racemosus* f. *racemosus*	222.81	CBS	0.062
*Mucor racemosus* f. *sphaerosporus*	115.08	CBS	0.054
*Mucor ramosissimus*	135.65	CBS	0.051
*Phycomyces nitens*	133	CRT	0.073
*Rhizomucor pusillus*	120586	CBS	0.044
*Rhizomucor pusillus*	120587	CBS	0.081
*Rhizopus arrhizus*	T14A	CRT	1.355
*Rhizopus arrhizus*	2601	NCPF	1.442
*Rhizopus arrhizus*	2634	NCPF	1.392
*Rhizopus arrhizus* var. *arrhizus*	112.07	CBS	1.395
*Rhizopus arrhizus* var. *arrhizus*	118614	CBS	1.365
*Rhizopus arrhizus* var. *delemar*	544.8	CBS	1.466
*Rhizopus arrhizus* var. *delemar*	607.68	CBS	1.622
*Rhizopus azygosporus*	357.93	CBS	0.010
*Rhizopus homothallicus*	336.62	CBS	0.030
*Rhizopus microsporus* var. *oligosporus*	tempeh starter (Raprima)	CRT	0.040
*Rhizopus microsporus* var. *rhizopodiformis*	102277	CBS	0.013
*Rhizopus schipperae*	138.95	CBS	0.030
*Rhizopus oryzae*	102659	CBS	1.369
*Rhizopus oryzae*	111233	CBS	1.407
*Rhizopus oryzae*	tempeh starter (Scot)	CRT	1.225
*Rhizopus stolonifer* var. *stolonifer*	389.95	CBS	0.020
*Scedosporium apiospermum*	117467	CBS	0.083
*Scedosporium aurantiacum*	121926	CBS	0.066
*Saksenaea vasiformis*	113.96	CBS	0.053
*Syncephalastrum racemosum*	155	CRT	0.061

^1^ CBS; Westerdijk Fungal Biodiversity Institute, The Netherlands. CRT; C. R. Thornton, University of Exeter, UK. NCPF; National Centre for Pathogenic Fungi, Public Health England, UK. ^2^ For ELISA using mAb KC9, mean absorbance values greater than the threshold value for test positivity (≥0.100) show antigen recognition; mean absorbance value less than the threshold value for test positivity are negative for antigen recognition.

## Data Availability

The data presented in this study are available on request from the corresponding author, but are not publicly available due to commercial confidentialities. Monoclonal antibody, KC9, and the LFD are available through ISCA Diagnostics Ltd.

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
