# Peer review of "Development of a Monoclonal Antibody and a Serodiagnostic Lateral-Flow Device Specific to Rhizopus arrhizus (Syn. R. oryzae), the Principal Global Agent of Mucormycosis in Humans"

_jof, 2022, doi:10.3390/jof8070756_

Round 1
Reviewer 1 Report
Davies and Thornton described the development of a lateral-flow device specific to Rhizopus arrhizus. This aim pursued by the authors could have a high clinical relevance, particularly for resource-limited settings, since at the present time the cultural detection of Mucorales is central for the diagnosis.
Expensive and time-consuming PCR methods support the diagnosis. Due to the increasing number of immunosuppressed patients, thanks to the rapid progress of hemato-oncological therapies, more and more people are at higher risk towards invasive fungal infection. In addition, in the wake of the COVID19 pandemic, increased opportunistic cases of mucormycosis have been described, particularly in developing countries. Resource scarcity is particularly prevalent here, so more cost-effective alternatives to the PCR kits currently commercially available could bring important advances. However, using the competitive LFD in which the relative intensity of the Test must be compared with the Control and determine a reduction of band intensity, is only possible with a reader, which limits its use in in developing countries. In developed countries other detection methods will be preferred. The experience from COVID-19 LFD tests that the intensity of the Test line may vary but is always positive would be a better approach. Not knowing the clinical relevance of the limit of detection of 50 ng/mL in human mucormycosis is also a drawback of this method. The objective is to detect the infection in an early and treatable stage.
My advice is to change the competitive LFD to a sandwich LFD test so that it has a lower LOD, to really save patients and to implement in developing countries. The hard work is already done and the mAb-KC9 is very promising.
The authors refer that they have developed an Apophysomyces-specific mAb 388 (JD4) and a pan-mucorales-specific mAb (TG11) for incorporation into a multiplex LFD 389 alongside KC9 to overcame the disadvantage of this LFD in detecting only R. arrhizus. This is a very interesting approach and them maybe the presence of the KC9 may be irrelevant. Why present this approach now ?
Introduction
Mucormycosis is a serious but rare fungal infection, particularly in people who have health problems or take medicines that lower the body’s ability to fight microorganisms. The authors should contextualize these infections in the spectrum of the fungal infections.
Line 44- “mucormycosis is associated with high mortality, please indicate numbers / specific manifestation of the disease.
Material and methods
Line 72- explain why fungi were grown under a 16 h fluorescent light regime.
Line 83- is the method for extracellular polysaccharides extraction described anywhere? A reference should be included or the method described.
Sentence 86-89 is difficult to understand. Rephrase.
Line 94- was the blocking performed before incubation with TCS?
Antibody Specificity Tests – were these tests performed with extracellular polysaccharides extracted by using the same protocol ?
Line 158- why were the plates dried ? was this procedure performed with extracellular polysaccharides from Rhizopus arrhizus ?
Line 188- “sample was mixed with 7.5GU of a 1.5 ug/mL solution of KC9-gold conjugate” If I understand KC9 was conjugated with HRP ?
Results
Line 225- “cell line KC9 was selected for further evaluation due to its strength of reaction in ELISA, and lack of cross-reactivity with other fungi in specificity tests” could these results be presented as absorbance values?
Epitope Characterisation- Could not find the S1A and S1B images
Section LFD
I do not understand the difference between the ELISA (Fig 2B) and western blot (Fig 2D) specificity-tests in this section compared with the same tests performed previously. If they are the same these figures should be in the previous section.
The analytical limit of detection (LOD) of this type of LFD will always be dependent of the amount of mAb KC9-conjugate. Explain how this could affect / improve the LOD.
The information on the legend of figure 3 regarding LOD should be in the main text so that readers can understand the score rationale.
Unfortunately, using a competitive LFD in which the relative intensity of the Test must be compared with the Control to determine a reduction of band intensity would be very difficult to implement. A change from a 8-6 (control) intensity to a 5 (LOD) visually is very difficult without the reader.
The clinical relevance of the limit of detection must be determined other wise the LFD may not be useful.
Author Response
Davies and Thornton described the development of a lateral-flow device specific to Rhizopus arrhizus. This aim pursued by the authors could have a high clinical relevance, particularly for resource-limited settings, since at the present time the cultural detection of Mucorales is central for the diagnosis.
Expensive and time-consuming PCR methods support the diagnosis. Due to the increasing number of immunosuppressed patients, thanks to the rapid progress of hemato-oncological therapies, more and more people are at higher risk towards invasive fungal infection. In addition, in the wake of the COVID19 pandemic, increased opportunistic cases of mucormycosis have been described, particularly in developing countries. Resource scarcity is particularly prevalent here, so more cost-effective alternatives to the PCR kits currently commercially available could bring important advances. However, using the competitive LFD in which the relative intensity of the Test must be compared with the Control and determine a reduction of band intensity, is only possible with a reader, which limits its use in in developing countries. In developed countries other detection methods will be preferred. The experience from COVID-19 LFD tests that the intensity of the Test line may vary but is always positive would be a better approach. Not knowing the clinical relevance of the limit of detection of 50 ng/mL in human mucormycosis is also a drawback of this method. The objective is to detect the infection in an early and treatable stage.
Rapid antigen test based on lateral-flow technology are ideally suited to resource limited settings. The test described here is portable, cheap, quick and simple compared to PCR, and does not require specialist training or access to well-equipped laboratories with expensive and sophisticated machinery. The small and portable Cube reader runs on batteries and provides a quick and simple means of determining test positivity based on arbitrary units of test and control line intensities or when used with internal thresholds for test positivity.
There is no difference in interpretation of test results in the two different LFD formats (Sandwich and Competitive). One looks for gain-of-signal (e.g. COVID-19 Sandwich LFD tests and Aspergillus Sandwich LFDs), the other looks for loss-of-signal (Competitive format - see lines 724 to 736 in the Discussion). Interpretation is not difficult to implement, especially when using a portable Cube reader. Both commercial Aspergillus Sandwich LFDs (OLM Diagnostics and IMMY) require use of a Cube reader despite both tests being readable by the naked eye. We have shown here that there is a direct correlation between loss of test line intensity and increased EPS concentration in the analyte, both with the Cube reader and with subjective appraisal using the naked eye (Figures 3C and 3D). Any loss of test line intensity compared to the control indicates a positive test, in the same way that appearance of a test line (either weak, moderate or strong) indicates a positive test result in a Sandwich format. From 2025, changes in IVD regulations mean that all commercially available CE-marked LFD tests will need to be accompanied by positive and negative controls regardless of whether they are Sandwich or Competitive formats. Clinical relevance of the LOD of the test is addressed below.
My advice is to change the competitive LFD to a sandwich LFD test so that it has a lower LOD, to really save patients and to implement in developing countries. The hard work is already done and the mAb-KC9 is very promising.
Our work has shown that extracellular antigens of mucoralean fungi appear to be very different to those of other filamentous fungi. Their species-specific extracellular polysaccharide antigens (and indeed genus-specific antigens) lack repeat epitopes that allow the development of sandwich assays. We have already provided an explanation for our choice of the Competitive format in lines 731 to 736, with a wider justification for Competitive lateral-flow assays in lines 724 to 736.
The authors refer that they have developed an Apophysomyces-specific mAb 388 (JD4) and a pan-mucorales-specific mAb (TG11) for incorporation into a multiplex LFD 389 alongside KC9 to overcame the disadvantage of this LFD in detecting only R. arrhizus. This is a very interesting approach and them maybe the presence of the KC9 may be irrelevant. Why present this approach now?
Yes, it is possible that KC9 may become irrelevant, but this is highly unlikely. The use of more than one biomarker adds significant power to diagnostic accuracy. This has been shown in the case of aspergillosis where the combined use of lateral-flow technology with PCR and/or with beta-D-glucan and galactomannan tests substantially improves diagnostic accuracy. Furthermore, both from an epidemiological perspective and from a treatment perspective, it is important to speciate the infecting organism as they differ in sensitivities to different antifungal drugs.
Introduction
Mucormycosis is a serious but rare fungal infection, particularly in people who have health problems or take medicines that lower the body’s ability to fight microorganisms. The authors should contextualize these infections in the spectrum of the fungal infections.
We have added an opening sentence to the Introduction (lines 31 to 33) to place mucormycosis in the context of other mould infections of humans, especially aspergillosis.
Line 44- “mucormycosis is associated with high mortality, please indicate numbers / specific manifestation of the disease.
Details of manifestations can be found on lines 36 to 42. We have added further detail to lines 48 to 52. We have also added the overall all-cause mortality rate of 54% to lines 53-54.
Material and methods
Line 72- explain why fungi were grown under a 16 h fluorescent light regime.
We have added the explanation to line 82.
Line 83- is the method for extracellular polysaccharides extraction described anywhere? A reference should be included or the method described.
No, the method is not described elsewhere. As stated in the methodology, this is a proprietary method developed by industry. To mitigate this, we have made available the KC9 antibody through the ISCA Diagnostics website (as stated on lines 780 to 781), enabling access to the antibody and test by the wider scientific community. Should others wish to use the KC9 antibody to identify antigenic preparations containing the EPS biomarker then they can do so using the detailed methodologies provided.
Sentence 86-89 is difficult to understand. Rephrase.
With respect, we do not find this confusing, but have made a minor modification to sentence construction.
Line 94- was the blocking performed before incubation with TCS?
No, no blocking was performed before incubation with TCS. We do not use blocking during the early screening stages in our laboratory as we find it can mask non-specific binding with certain mAbs, creating difficulties further downstream in development. We prefer to identify highly-specific antibodies at the start of the process which demonstrate no non-specific binding with raw, unblocked TCS.
Antibody Specificity Tests – were these tests performed with extracellular polysaccharides extracted by using the same protocol?
We have modified the paper to better explain this.
Line 158- why were the plates dried? was this procedure performed with extracellular polysaccharides from Rhizopus arrhizus?
This is standard operating procedure in our laboratory and many other diagnostic laboratories.
Line 188- “sample was mixed with 7.5GU of a 1.5 ug/mL solution of KC9-gold conjugate” If I understand KC9 was conjugated with HRP?
No, as stated it is a gold conjugate. We have modified the sentence (lines 472 to 473) to make this clearer.
Results
Line 225- “cell line KC9 was selected for further evaluation due to its strength of reaction in ELISA, and lack of cross-reactivity with other fungi in specificity tests” could these results be presented as absorbance values?
Yes, we now present these as absorbance values in Table 1.
Epitope Characterisation - Could not find the S2A and S2B images.
These are present in the Supplementary Figures file as Figures S2A and S2B. These were Figures S1A and S2B in the original version.
Section LFD
I do not understand the difference between the ELISA (Fig 2B) and western blot (Fig 2D) specificity-tests in this section compared with the same tests performed previously. If they are the same these figures should be in the previous section.
They are different. Table 1 (legend now modified) shows the results of ELISA tests for KC9 obtained during initial screens of hybridoma cells lines. The western blots in Fig. 2C and 2D provide a visual appraisal of KC9 specificity based on the presence or absence of the ~15 kDa biomarker in different Rhizopus species. The western blots using the pAb (Figs. 2A and 2B) show that while antibody-reactive antigens are present in all of the culture filtrates, the KC9 mAb is specific for an ~15 kDa antigen of Rhizopus arrhizus/oryzae.
The analytical limit of detection (LOD) of this type of LFD will always be dependent of the amount of mAb KC9-conjugate. Explain how this could affect / improve the LOD.
The amount of KC9 conjugate has been titered to provide the greatest test sensitivities across the different analytes (running buffer, serum, and BALf).
The information on the legend of figure 3 regarding LOD should be in the main text so that readers can understand the score rationale.
We have amended the legend and main text as suggested.
Unfortunately, using a competitive LFD in which the relative intensity of the Test must be compared with the Control to determine a reduction of band intensity would be very difficult to implement. A change from a 8-6 (control) intensity to a 5 (LOD) visually is very difficult without the reader.
The clinical relevance of the limit of detection must be determined otherwise the LFD may not be useful.
There is no difference in interpretation of test results in the two different LFD formats (Sandwich and Competitive). One looks for gain-of-signal, the other looks for loss-of-signal (see lines see lines 724 to 736 in the Discussion). We disagree that the interpretation is difficult to implement, especially when using a portable Cube reader. Both Aspergillus Sandwich LFDs (OLM Diagnostics and IMMY) require use of a Cube reader despite both tests being readable by the naked eye. This is because of false positive results, especially in the case of the IMMY test that displays non-specific binding with control (negative) samples. We have shown here that there is a direct correlation between loss of test line intensity and increased EPS concentration in the analyte both with the Cube reader and with subjective appraisal using the naked eye (Figures 3C and 3D). Any loss of test line intensity compared to the control indicates a positive test, in the same way that appearance of a test line (either weak, moderate or strong) indicates a positive test result in a Sandwich format. From 2025, changes in IVD regulations mean that all commercially available CE-marked LFD tests will need to be accompanied by positive and negative controls regardless of whether they are Sandwich or Competitive formats.
Yes, we agree that the clinical relevances of the limits of detection with running buffer, serum and BALf must be determined. We state this clearly in the Discussion (lines 749 to 758). However, this is beyond the scope of the current paper. This aim of this paper is to describe the first ever rapid antigen test for the principal agent of mucormycosis (R. arrhizus) using a novel antigenic biomarker, and demonstration of its applicability to human serum and BALf fluids. Testing of patient samples will of course be undertaken subsequently and reported in a follow-up paper.
Reviewer 2 Report
This manuscript makes a relevant contribution to the diagnosis of Rhizopus arrhizus by developing a competitive lateral-flow device (LFD) based on the monoclonal antibody KC9 against an extracellular polysaccharide (EPS) antigen secreted by R. arrhizus, but not by other Mucorales. Despite these achievements are important for the diagnosis and study of mucormycosis, there are several concerns, described below, about the work.
- - Results section makes a concise description of the results, in some cases without explanation, like in section 3.4.2. Authors should revise the whole Results section to add appropriate descriptions and explanations when required.
- - As the authors indicated, the main problem of the KC9 antibody is that it detects only EPS from R. arrhizus, but there many species and genera that cause mucormycosis, and they represent a significant proportion of mucormycosis cases. This fact reduces the usefulness of this antibody for diagnosis.
- - The second most important concern is about its application in real diagnosis. The authors should validate the device in patients infected with the fungus. There are some important questions related to this point. The first and most important one is about the production of the EPS during the infection. It should be confirmed. The second one is about the sensitivity. Authors should check the LFD with infected patients because sensitivity seems to be low.
- - No clear description was found about the origin of pAb SK0078.
Minor points:
- - Lane 11-15. Sentence too long.
- - Lane 16-17. Molecular techniques of diagnosis deserve to be mentioned. Their use is continuously increasing.
- - Lane 83-84. This method should be described.
- - Figure 2. The symbols for R. oryzae CBS111233 in figure and inset are different.
Author Response
This manuscript makes a relevant contribution to the diagnosis of Rhizopus arrhizus by developing a competitive lateral-flow device (LFD) based on the monoclonal antibody KC9 against an extracellular polysaccharide (EPS) antigen secreted by R. arrhizus, but not by other Mucorales. Despite these achievements are important for the diagnosis and study of mucormycosis, there are several concerns, described below, about the work.
Results section makes a concise description of the results, in some cases without explanation, like in section 3.4.2. Authors should revise the whole Results section to add appropriate descriptions and explanations when required.
We have modified the Results section to reflect this comment.
As the authors indicated, the main problem of the KC9 antibody is that it detects only EPS from R. arrhizus, but there many species and genera that cause mucormycosis, and they represent a significant proportion of mucormycosis cases. This fact reduces the usefulness of this antibody for diagnosis.
We disagree wholeheartedly with this comment. As stated in the Abstract, Introduction and Discussion sections, and supported by numerous published articles (many of which are cited here), Rhizopus arrhizus is the principal cause of mucormycosis globally. Yes, there are other species that cause the disease and we make this clear in the Discussion (lines 762 to 768). That is why follow-up studies aim to develop a complementary pan-mucorales multiplex test (see lines 762 to 768).
The second most important concern is about its application in real diagnosis. The authors should validate the device in patients infected with the fungus. There are some important questions related to this point. The first and most important one is about the production of the EPS during the infection. It should be confirmed. The second one is about the sensitivity. Authors should check the LFD with infected patients because sensitivity seems to be low.
Yes, we agree, and address the issue of patient sample testing already. However, this is beyond the scope of the current paper. This aim of this paper is to describe the first ever rapid antigen test for the principal agent of mucormycosis (R. arrhizus) using a novel antigenic biomarker, and demonstration of its applicability to human serum and BALf fluids. Testing of patient samples will of course be undertaken subsequently and reported in a follow-up paper.
The EPS antigen is secreted into culture fluids during active growth (see Supplementary Figures S1A and S1B, lines 528 to 531 (Section 3.2 of the Results) and lines 690 to 693 (Discussion). Given its secretion during active hyphal growth (Figure S1B), there is no reason to believe that the antigen won’t be secreted during invasive growth in a human.
We question the statement ‘sensitivity seems to be low’. The levels of sensitivity are similar to those found with Sandwich LFDs for aspergillosis detection e.g. the AspLFD test developed by the corresponding author and sold commercially by OLM Diagnostics (~50ng/mL for running buffer, ~100ng/mL for BALf and an approximately 10-fold loss of sensitivity for serum). We have alluded to this in lines 750 and 755, referring to two references where similar levels of sensitivity of sandwich LFDs are shown (see references 1 and new reference 104).
No clear description was found about the origin of pAb SK0078.
This is already described under Section 2.5 (lines 375 to 378) with its purification and enzyme conjugation then further described in Section 2.6 (lines 391 to 400).
Minor points:
Lane 11-15. Sentence too long.
We have modified the sentence construction.
Lane 16-17. Molecular techniques of diagnosis deserve to be mentioned. Their use is continuously increasing.
With respect, we believe this is adequately addressed in the subsequent Introduction and Discussion sections and does not need to be elaborated in the Abstract.
Lane 83-84. This method should be described.
As stated in the methodology, this is a proprietary method developed by industry. To mitigate this, we have made available the KC9 antibody through the ISCA Diagnostics website (as stated on lines 780 to 781), enabling access to the antibody and test by the wider scientific community. Should others wish to use the KC9 antibody to identify antigenic preparations containing the EPS biomarker then they can do so using the detailed methodologies provided.
Figure 2. The symbols for R. oryzae CBS111233 in figure and inset are different.
This has now been corrected.
Round 2
Reviewer 1 Report
The authors addressed all major questions and have adapted the manuscript accordingly.
Reviewer 2 Report
The authors addressed all the concerns.